# Using Soft Sensors as a Basis of an Innovative Architecture for Operation Planning and Quality Evaluation in Agricultural Sprayers [note 1]

**DOI:** 10.3390/s21041269

**Published:** 2021-02-10

**Authors:** Elmer A. G. Peñaloza, Vilma A. Oliveira, Paulo E. Cruvinel

**Affiliations:** 1Engineering Center, Federal University of Pelotas, Rua Benjamin Constant, n° 989, Porto, Pelotas 96010-020, RS, Brazil; 2Department of Electrical and Computer Engineering, University of São Paulo, São Carlos, Av. Trabalhador São Carlense, n° 400, São Carlos 13566-590, SP, Brazil; voliveira@usp.br; 3Embrapa Instrumentation, Brazilian Agricultural Research Corporation, Rua XV de Novembro, n° 1.452, São Carlos 13560-970, SP, Brazil; paulo.cruvinel@embrapa.br

**Keywords:** soft-sensor design, inferential sensors, quality of application, principal component analysis, K-nearest neighbor, agricultural sprayers

## Abstract

One of the major problems facing humanity in the coming decades is the production of food on a large scale. The production of large quantities of food must be conducted in a sustainable and responsible manner for nature and humans. In this sense, the appropriate application of agricultural pesticides plays a fundamental role since pesticide application in a qualified manner reduces human and environmental risks as well as the costs of food production. Evaluation of the quality of application using sprayers is an important issue, and several quality descriptors related to the average diameter and distribution of droplets are used. This paper describes the construction of a data-driven soft sensor using the parametric principal component regression (PCR) method based on principal component analysis (PCA), which works in two configurations: with the input being the operating conditions of the agricultural boom sprayers and its outputs being the prediction of the quality descriptors of spraying, and vice versa. The soft sensor provides, in one configuration, estimates of the quality of pesticide application at a certain time and, in the other, estimates of the appropriate sprayer-operating conditions, which can be used for control and optimization of the processes in pesticide application. Full cone nozzles are used to illustrate a practical application as well as to validate the usefulness of the soft sensor designed with the PCR method. The selection of historical data, exploration, and filtering of data, and the structure and validation of the soft sensor are presented. For comparison purposes, the results with the well-known nonparametric k-Nearest Neighbor (k−NN) regression method are presented. The results of this research reveal the usefulness of soft sensors in the application of agricultural pesticides and as a knowledge base to assist in agricultural decision-making.

## 1. Introduction

With the rise in the data-processing capacity and the speed of calculation in the new generation of processors embedded in small devices, it is easier to create virtual instruments based on information and models obtained from the production process. Therefore, mathematical models can be used to represent the variables that cannot be measured in a process based on the variables that are available and can be easily measured with instruments. Soft sensors are computer programs established on models that are used for estimating unmeasurable variables from production processes; specifically, they are based on estimation and prediction techniques that use a priori information collected using sensors and mathematical models that describe physical processes.

A soft-sensor-based approach is used in cases when sensors (hardware) are unavailable, when their implementation is difficult and incurs high costs, or when no instruments can measure the variable of interest [1]. The application of soft sensors are generally divided into three categories: monitoring processes, process control, and offline assistance for process operations [2].

In the literature, there are several successful applications of soft-sensors in production processes. In 1995, an inference estimator based on fuzzy logic to measure and control the purity of propylene from a high-purity distillation column was designed [3]. Here, estimation was made by adopting the distillation process model and by using it as the knowledge base for training the input and output data of the plant for specific situations. Using this approach, the authors were able to accurately model nonlinear systems with an online learning capability. In 1998, soft sensors were used to estimate the size of particles in a grinding plant, where sensors were unavailable [4]. The authors in [4] used an autoregressive moving average model (ARMAX) as a soft sensor to estimate and test the predictive capacity. Then, in 2007, a soft sensor to detect nitrogen oxide (NO${}_{x}$) emissions produced by a cement kiln system was designed [5]. The authors in [5] used robust regression techniques to derive an inferential model, making estimation possible using dynamic least squares. In 2016, a soft-sensor approach to predict and monitor indoor air quality in the Seoul metro system was used [6]. The authors in [6] used the just-in-time (JIT) learning technique to model the nonlinear process based on two local models of prediction: the linear partial least squares (PLS) and the nonlinear least squares support vector regression (LSSVR).

Recently, studies have emerged with relevant use of soft sensors, an example of which is the approach appearing in [7] to determine physical properties of different materials based on the historical of spectroscopic readings of the samples tested a priori. The authors in [7] proposed the use of intelligent models to determine the correlation between different wavelengths and to determine which variables have more statistical weight in a whole spectrum. The methodology used by these authors are based on the statistics pattern analysis (SPA), which offers good results by reducing the complexity of models and by improving the estimation performance. In addition, in [8], the soft-sensor approach was applied to estimate the dissolved oxygen level in a hydraulic recirculation system used for aquaculture using recurrent neural networks (RNN).

Another recent application of the soft-sensor approach was carried out in [9] for real-time estimate and monitoring of phosphates and soluble chemical oxygen demand (COD) concentrations in the anaerobic chambers of a multistage moving bed biofilm reactor (MBBR) configuration. The soft sensor was developed from an extended Kalman filter applied to a reduced-order nutrient removal analytical model. The validation of the results demonstrates the success of the soft-sensor approach to estimate these types of variables. On the other hand, there is currently concern in improving existing technologies and in developing new base methodologies for the construction of soft sensors to predict quality in industrial processes. An example of this is the development of a new soft sensor approach based on a multichannel convolutional neural network (MCNN) recently proposed in [10] showing acceptable results in estimating quality variables of the debutanizer column and hydrocracking industrial processes.

The use of statistical approaches for the construction of a soft sensor is widely studied in the literature; an example of this is the research developed in [11] in which the authors developed a methodology for the construction of a soft-sensor based on principal component analysis (PCA) for the detection of sensor failures. The soft-sensor model was built based on historical data taken from an actual nuclear power plant. The authors used and compared two models, one based on an improved PCA model and the other based on a cyclical PCA model. Both soft sensors can quickly detect the occurrence of multiple sensor faults and can successfully isolate these faulty sensors of the process. In the same line, in [12], an approach based on PCA to develop a soft sensor to perform sensor failure detection in a real water source heat pump air-conditioning system was used. The PCA statistical approach is used in conjunction with a *k*-means clustering, to optimize the classification prepossessing of both training and test data. The results obtained by the researchers showed that the use of these methodologies offers a strong detection capability when random faults are introduced to the sensor in the execution of the process.

The construction of a soft sensor starts with knowledge of the process and the relationship among the relevant variables. Therefore, it is important to recognize all variables involved in the process to identify the variables to be sensed and the variables to be estimated or predicted. The conception and construction of a data-driven soft sensor have five main pillars: collection and selection of historical process data, detection of outliers and data filtering, selection of the model structure, estimation, and validation of the model [1]. Therefore, these five pillars must be executed sequentially to obtain a soft sensor with a high degree of accuracy.

As the global population increases, the need to produce more food force agricultural techniques constantly evolves.The development of new technologies for the production of inputs, pesticides, and agricultural machines such as tractors and sprayers, and genetic engineering have made it possible to increase agricultural production and to reduce the environmental impact. Among the activities of crop management, one of the most expensive is spraying pesticides. Spraying is the application of a liquid in the form of small particles on a surface. These particles are called drops or droplets.

An efficient spraying application is based on the following factors: efficiency of the spraying application, quality of the applied chemical, climatic conditions, and biological characteristics of the pest [13,14]. Among these factors, spraying quality is one of the most important and precision agriculture based on automation and control plays an important role. The knowledge of the size, distribution, and process of droplet formation are essential for the successful pulverization of pesticides [15]. These have influences on the drift, evaporation of products, penetration capability inside the canopy of crops, and deposition on phytosanitary treatment targets [16]. Also, the application speed as well as the nozzle position in the application boom, may affect the droplet size.

Because agricultural crops vary in height as they grow and an agricultural sprayer is used on different crops in a farm, the sprayer boom height must be accurate to ensure that crops receive proper amounts of liquids dispensed. Furthermore, current advanced sprayers generally include additional sets of sensors, which are useful for precision spraying management. A set of sensors have been used to help the operators in the calibration of the engine temperature, monitoring of flow and pressure of the pesticide hydraulic pump, and other variables required in the spraying process.

However, it is still a challenge to measure and control all the variables required to guarantee the spraying quality and to obtain a complete characterization of the spraying system during operation. Therefore, to improve the performance of such processes, soft sensors are used to estimate the values of important variables that cannot be obtained through traditional measurements.

In this paper, we present an innovative soft-sensor architecture to improve operation planning and quality evaluation of the agricultural processes in pesticide application based on statistical models and statistical pattern analysis. The main focus of this study is the prediction of quality descriptors of the application as a function of the operating conditions of agricultural sprayers as well as the operational planning of the agricultural processes. Larger pressure nozzle ranges and spray boom nozzle positions are considered. The operating conditions considered are thus related to droplets sizes and nozzle orifice diameters in order to select the best nozzle type for each operating condition, and this would be useful to automatically perform individual control of each nozzle.

## 2. Materials and Methods

This section begins by presenting the boom sprayer quality descriptors used including the Sauter mean diameter, which provides information on the uniformity of the droplet spectrum. Following, the regression models, the conception of the soft-sensor in terms of inputs and outputs, the experimental setup, data collection, and the validation methods are presented.

### 2.1. Droplet Size and Distribution

In the literature, a function of the distribution of instantaneous diameters, which are typically used to describe sprayers, is widely used [17,18]. Such a function provides information on the number of drops having a certain average diameter and the distribution of these diameters for a particular spray application. In general, the mean diameter denoted $D_{cd}$ represents the characteristics of the spraying. Let $N_{i}$ and $D_{i}$ be the number of drops in the size range *i* and the mean diameter of size range *i*, respectively. The diameter function $D_{cd}$ is discretely calculated using the following equation:(1)$$D_{cd}={\left(\frac{\Sigma N_{i}D_{i}^{c}}{\Sigma N_{i}D_{i}^{d}}\right)}^{\frac{1}{\left(c-d\right)}}$$
where *c* and *d* are positive integers, $D_{cd}$ is given in the unit of diameter, and *i* denotes the range of size considered.

The volumetric median diameter (VMD) is important in the characterization of spraying and is widely used in agricultural spraying [19,20,21]. This diameter is calculated by substituting c=3 and d=0 into (Equation 1) to obtain the following expression:(2)$$VMD={\left(\frac{\Sigma N_{i}D_{i}^{3}}{\Sigma N_{i}}\right)}^{\frac{1}{3}}$$
where VMD represents the median of droplet volumes in the spray [22], that is, VMD is the diameter of a droplet in spectrum, which divides the volume into two equal parts: one consisting of droplets with smaller diameters and the other with droplets of larger diameters.

On the other hand, the Sauter mean diameter (SMD) describes the relationship between the total droplet volume in a spray and the total surface area of the droplets [22]. This mean diameter is calculated by substituting c=3 and d=2 into (Equation 1) as follows:(3)$$SMD=\frac{\Sigma N_{i}D_{i}^{3}}{\Sigma N_{i}D_{i}^{2}}$$

The spraying median diameters provide information regarding the volume as a function of the frequency of formed droplet sizes, but this information is insufficient when analyzing the uniformity of spraying. Therefore, it is necessary to consider some representative diameters. One of these is the diameter of droplets such that 10 % of the total volume of liquid is in drops of smaller diameter, named $D_{0.1}$, and the other is the diameter of droplets such that 90% of the total volume of liquid is in drops of smaller diameter, named $D_{0.9}$.

The representative diameters are also used to characterize the relative amplitude. The relative amplitude RA is defined as follows:(4)$$RA=\frac{D_{0.9}-D_{0.1}}{VMD}.$$

This parameter, quantifies the range of sizes containing 80% of the spray volume and is a nondimensional comparative index of the droplets that compose the spray. In addition, the relative amplitude provides an indication of the difference in droplet sizes per VMD. Therefore, the greater the relative amplitude, the greater the degree of heterogeneity of the spray spectrum [13].

In relation to droplet size and distribution, it is important to observe that other factors exist, which are implicit in hydrodynamic and aerodynamic processes owing to the formation and rupture of liquid jets. However, the current theories are insufficient to describe the formation of the size and distribution of droplets in the liquid spraying process. Therefore, empirical correlations are used to predict the droplet size and distribution. A common empirical function used to describe the distribution of droplet diameters is the Rosin–Rammler empirical function. This function relates the total volume fraction to the droplet diameters. The Rosin–Rammler function is expressed as follows:(5)$$1-Q=\exp{\left(-D/R\right)}^{q}$$
where *D* (μm) is a given diameter, *Q* is the fraction of the total volume of drops with diameter less than *D*, and *q* and *R* are the parameters of the Rosin–Rammler distribution with *q* as the drop size and *R* as its diameter. The exponent *q* provides the measure of the spread of drop sizes. Larger values of *q* lead to greater spray uniformity. The distribution parameter *q* can thus be computed using the experimental median diameters ($D_{0.9}$ and VMD) as follows:(6)$$\frac{D_{0.9}}{VMD}={\left(3.32\right)}^{\left(1/q\right)}.$$

The Rosin–Rammler distribution allows for the extrapolation of data in the range of very fine drops, where measurements are more difficult and less precise [23]. The advantage of using known distribution functions is that we can easily find mathematical relationships between different diameters; for example, in the Rosin–Rammler distribution, the relations are made as a function of the drop size distribution parameter *q*. Therefore, the rate between *SMD* and *VMD* can be related as follows [24]:(7)$$\frac{SMD}{VMD}={\left(0.693\right)}^{\left(1/q\right)}\Gamma \left(1-\frac{1}{q}\right)$$
where Γ is the Gamma function.

In this work, the average diameters *VMD* (Equation 2) and *SMD* (Equation 3), the characteristic diameters $D_{0.1}$ and $D_{0.9}$, the application rate (AR), the covered area (CA) by spraying, and the uniformity index that is represented by the relative amplitude RA (Equation 4) are used as the descriptors of quality in the agricultural spraying process.

### 2.2. Regression Models

In what follows, the regression models based on PCR by PCA analysis as well as the nonparametric regression method, based on the *k* Nearest Neighbor (*k*-NN) used for comparison are summarized for easy reference.

#### 2.2.1. Principal Component Analysis

A regression model is commonly used to represent experimental data. The regression model coefficients are obtained using principal components (PCs). The main idea here is to reduce the dimension of the data set while keeping the variation of the original data.

Consider a set of observations $\left\{x_{n}\right\}$, where n=1,⋯,N and $x_{n}$ is a Euclidian variable with dimensionality *D*. To obtain the PCs, the projection of observations onto a space with dimensionality M<D is performed. To present this formulation, the simplest one-dimensional space case (M=1) ) is used, i.e., the projection of data is in the one-dimensional space [25]. The mean of observations is calculated as follows:(8)$$\bar{x}=\frac{1}{N}\sum_{n=1}^{N}x_{n}.$$

The covariance matrix S is defined using the following expression:(9)$$S=\frac{1}{N}\sum_{n=1}^{N}\left(x_{n}-\bar{x}\right){\left(x_{n}-\bar{x}\right)}^{T}.$$

Let a D-dimensional vector $u_{1}$ be the direction of this space, which is chosen such that $u_{1}^{T}u_{1}=1$. Each vector; $x_{n}$ is then projected onto the scalar value $u_{1}^{T}x_{n}$, and the idea is to maximize the variance of the projected data in relation to the vector $u_{1}$. The variance of the projected data is given by
(10)$$\frac{1}{N}\sum_{n=1}^{N}{\left(u_{1}^{T}x_{n}-u_{1}^{T}\bar{x}\right)}^{2}=u_{1}^{T}Su_{1}.$$

To prevent $∥u_{1}∥\to \infty$, the maximization of the projected variance must be constrained. This constraint comes from the normalization condition, $u_{1}^{T}u_{1}=1$. To comply with this constraint, a Lagrange multiplier $\lambda_{1}$ is introduced [25]:(11)$$u_{1}^{T}Su_{1}+\lambda_{1}\left(1-u_{1}^{T}u_{1}\right).$$

Therefore, taking the derivative of (Equation 11) in relation to $u_{1}$ and equating it to zero, the following solution is obtained:(12)$$u_{1}^{T}Su_{1}=\lambda_{1}.$$

Therefore, $u_{1}$ is an eigenvector of the covariance matrix S, and the variance is maximized when the set $x_{1}$ is equal to the eigenvector with the largest corresponding eigenvalue $\lambda_{1}$. This eigenvector is known as the first PC [25]. The measure of contribution of certain eigenvector is contained in the corresponding eigenvalue.

Consider an M-dimensional space projection. The optimal linear solution obtained through maximizing the variance of projected data is given using *m* eigenvectors, $u_{1},\cdots ,u_{m}$ of the covariance matrix S that correspond to the *m* largest eigenvalues, $\lambda_{1},\cdots ,\lambda_{m}$, respectively. The *m* eigenvectors are the PCs and are ordered such that the first components keep most of the variation present in the original data or variables [26].

The standardization of data is typically performed when original variables are measured in different units or have significant variability, as is the case of quality descriptors. When calculating PCs, a linear rescaling must be made separately from each individual variable such that each variable has zero mean and unit variance.

#### 2.2.2. Principal Components in Regression Models

To establish PCs as a basis for modeling, we first define the (n×p) matrix X, which consists of *n* observations of *p* predictor variables with the (*i*,*j*)th element being the value of the *j*th predictor variable for its *i*th observation. Accordingly, the corresponding standard regression model is defined as follows:(13)y=Xβ+ϵ
where *y* is the vector of *n* observations of the dependent variable that are measured and are centred about their mean; β is the vector of *p* regression coefficients; and ϵ is the vector of error terms, where the elements of ϵ are independent, having the same variance $\sigma^{2}$. In addition, in a matrix form, PCs are the columns of the matrix Z, which is defined as Z=XA, where the (*i*, *k*)th element of Z is the PC for the *i*th observation, and A is a (p×p) matrix for which the *k*th column is the *k*th eigenvector of $X^{\prime }X$. The idea is to use PCs instead of the original observations in the regression model. Therefore, the concept of orthogonality of the eigenvector matrix is used. Since matrix A is orthogonal, Xβ is rewritten as $Z\gamma ={\mathrm{XAA}}^{\prime }\beta$, where $\gamma =A^{\prime }\beta$. Then, (Equation 13) can be rewritten as follows [26]:(14)y=Zγ+ϵ.

The following reduced model is used:(15)$$y=Z_{m}\gamma_{m}+\epsilon_{m}$$
where $\gamma_{m}$ is a vector with *m* elements, which are a subset of the elements of γ; $Z_{m}$ is an (n×m) matrix for which the columns are the corresponding subset of columns of Z; and $\epsilon_{m}$ is an appropriate error term. An estimate of β is found using $\hat{\beta }=A\hat{\gamma }$. The vector $\hat{\gamma }$ is calculated as $\hat{\gamma }={\left(Z^{\prime }Z\right)}^{-1}Z^{\prime }y$. Finally, the prediction of the interest variables is calculated using the following:(16)$$\hat{y}=Z_{m}\hat{\gamma_{m}}.$$

In what follows, for easy reference, a description of the *k*-NN estimates based on regression used for comparison with the PCA results is given.

#### 2.2.3. *k*-NN Regression

The *k* nearest neighbor is a nonparametric learning method. This method is based on distances, that is, compare new distances with distances already seen and stored in a previous training. Thus, the method searches the nearest neighbors of a distance using metrics. The most common metric used to evaluate the nearest neighbors of a point or distance is the Euclidean distance. Let *e* be a distance or observation which can be described by the characteristics vector $\left[a\left(e\right),a_{2}\left(e\right),\cdots ,a_{n}\left(e\right)\right]$, where $a_{r}\left(e\right)$ denotes the value of the *r*th attribute of the distance *e* [27]. Then, the Euclidean distance of $e_{i}$ and $e_{j}$ is defined as follows:(17)$$d\left(e_{i},e_{j}\right)=\sqrt{\sum_{r=1}^{N}{\left(a_{r}\left(e_{i}\right)-a_{r}\left(e_{j}\right)\right)}^{2}.}$$

Considering a continuous target function of the form $f:{ℜ}^{n}\to ℜ$ and a query or observation value $e_{q}$, an estimate of the value of the target function must be found, which should be the most nearest value of *f* among the *k* training examples nearest to $e_{q}$. The estimate $\hat{f}\left(e_{q}\right)$ of the target function of the nearest *k* values is defined as $\hat{f}\left(e_{q}\right)=\sum_{i}^{k}f\left(e_{i}\right)/k$, where k>0 which gives the mean value of the *k* nearest training examples.

An improvement in the *k*-NN algorithm is to weigh the contribution of each neighbor according to the distance to the query point $e_{q}$, thus giving a greater weight to the neighbors that are closer. Let $\omega_{i}$ be the weight assigned to a training distance $e_{i}$ defined in terms of the distance as $\omega_{i}=1/d{\left(e_{q},e_{i}\right)}^{2}$. The denominator of $\omega_{i}$ is zero when the query observation $e_{q}$ is equal to one of the training distances $e_{i}$, and in this case, the estimated value $\hat{f}\left(e_{q}\right)$ is set to $f(e_{i})$. Placing the weighting of the distances, to find the value of the target function $\hat{f}\left(e_{q}\right)$, the following expression is obtained:(18)$$\hat{f}\left(e_{q}\right)=\frac{\sum_{i}^{k}\omega_{i}f\left(e_{i}\right)}{\sum_{i}^{k}\omega_{i}}.$$

Taking the weighted average of the closest neighbors to the query point $e_{q}$ helps to smooth the impact of isolated noisy training samples.

### 2.3. Soft-Sensor Design

In this study, a soft sensor was developed to monitor processes, specifically, as a predictor of process quality descriptors (PPQD) and as an operational process planner (OPP). Two regression methods are used as the basis for the construction of the soft sensor: the first is based on a regression model via the covariance of the PCs (PC regression), and the second method is based on the mean value of the distance of the *k*-NN. The sequence of the development stages for the construction of the soft sensor for each method is explained in Algorithm 1. The execution of the algorithm begins with the choice of the method that will be the basis of construction of the soft sensor through the Boolean variable *B* in the initial condition structure (if). The programming of the PC regression algorithm and the *k*-NN algorithm were conducted in MATLAB-MathWorks^®^.

The steps for the construction of a model based on PC regression are summarized in Algorithm 1. This algorithm is divided into two procedures. The first procedure (REGRESSION) is the construction of the PC regression model. This procedure has a data matrix denoted X, the matrix of eigenvectors A, the matrix of scores of PCs Z, and the vector of data required for the model *y* as entries. In addition, this procedure returns the value of the regression coefficients $\hat{\gamma }$ and the predicted value $\hat{y}$. Therefore, the regression model is delivered based on PCs. The main function of the second procedure is to estimate output values for the new observed data (NEWOBSERV). Then, the procedure receives a vector containing new observations $x_{new}$, and the regression coefficients based on PCs $\hat{\gamma }$, and A. Here, a new score matrix $Z_{new}$, a new vector of values ${\hat{x}}_{new}$, and new observed data are estimated.

To compute the regression coefficients $\hat{\gamma }$, in Algorithm 1, the *p* columns of matrix X are the predictor variables, the 7 quality descriptors in the case of the PPQD soft-sensor and the 4 operating conditions in the case of the OPP soft sensor and the *n* lines of the matrix X are the observations obtained from the interpolation carried out on the experimental values, which are n=1000 in both cases. The column vector *y* contains the n=1000 training values of the variables that the soft-sensor delivers as output.

The procedures used to construct the soft-sensor based on the *k*-NN regression method is also shown in Algorithm 1. As inputs, the algorithm requires an attribute vector *x*, a vector containing the values of the target function *y*, and finally, the vector of query point $x_{q}$. As an output, the algorithm delivers an estimate of the values of the function $\hat{y}$ (variable to estimate), which is based on the query point vector $x_{q}$. The steps of Algorithm 1 relative to the *k*-NN estimates are composed of five main procedures.

The first procedure (DISTANCE) calculates the Euclidean distances between each of the components of the query point vector $x_{q}$ and the data stored in the attribute vector *x*. This procedure returns a vector of Euclidean distances. The second procedure in the algorithm (SORT) is responsible for sorting the Euclidean distances in an ascending order. Thus, this procedure returns an ordered pair containing the index and the value of the corresponding distance.

The third procedure (SEARCH) finds the nearest *k* neighbors of the query $x_{q}$ by the use of the ordered indexes. This procedure returns as output the set $T^{NN}$ of neighbors closest to the query vector. The forth procedure (WEIGHT) is responsible for weighting the distances found for the nearest neighbors. The procedure gives the set of weights corresponding to the inverse of the distance of the neighbors. Finally, the fifth procedure (ESTIMATE) assigns a majority weighted voting class attribute or label $\hat{y}$ to the query $x_{q}$. Then, this procedure returns the estimate of the attribute.

### 2.4. Soft-Sensor Architecture

The block diagram illustrated in Figure 1a explains the architecture of the soft sensor. The execution begins with the choice of the type of information that must be offered by the soft sensor, i.e., the predictor of the quality descriptors of the spraying process or the planner of the best spraying operating conditions that must be adjusted in the machinery. In the block diagram shown in Figure 1b, the control loop of the sprayer system is shown. In this loop, the input is the pressure reference $\Delta P_{ref}$. As the pressure and flow are related to each other, the regulated variable could be the flow $Q_{p}$ of the hydraulic sprayer system. The outputs of this loop are the measurements of the actual pressure denoted as ΔP.
**Algorithm 1:** Regression estimator
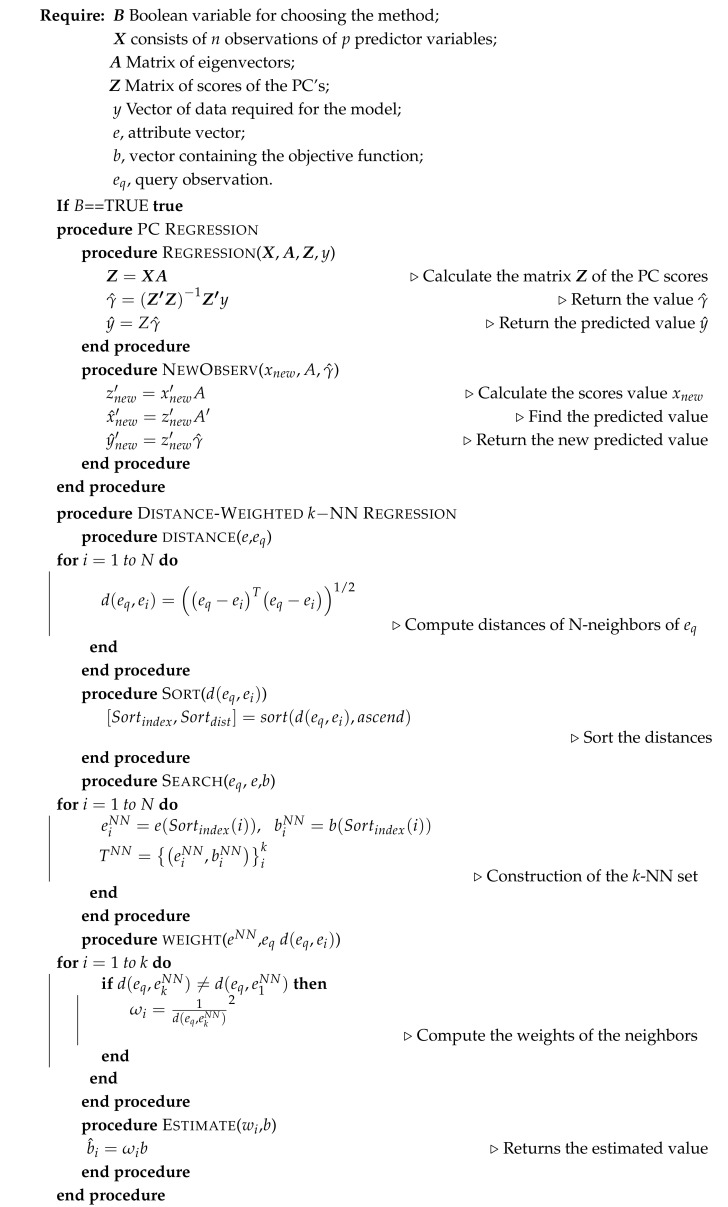


To select the inputs of the soft sensor, the studies were conducted with data obtained with the software DropScope^®^ for several operating conditions to register the most relevant drop patterns. In order to obtain the relationship among the variables, a simple PCA analysis was implemented and the inner product of the eigenvectors of the covariance matrix S was used. The vector angles have a representation in terms of linear correlation. Angles close to zero degrees represent a high positive correlation of the variables but angles close to $90^{\circ }$ indicate that the variables are independent, whereas angles close to $180^{\circ }$ indicate a high negative correlation. In Figure 2, the obtained PCA biplot are shown. From the PCA biplot analysis, the input operating conditions were selected as $O=[V_{p}\;d_{0}\;\Delta P\;Q_{p}]$, with $V_{p}$ as the speed of the application sprayer, $d_{0}$ as the diameter of the discharge orifice of the nozzle, ΔP and $Q_{p}$ as already defined, and the quality descriptor vector as Q = [D${}_{0.1}$ SMD VMD D${}_{0.9}$ RA CA AR].

Therefore, as a predictor PPQD, the operating conditions (red box) given by vector *O* are the inputs of the soft sensor. The operating conditions given by their samples $\{O_{i}\},i=1,\cdots ,N$ were used to mount the matrix X data in Algorithm 1. The variables that the soft sensor deliver as outputs are the predictions of a quality descriptor vector chosen as vector *Q*. As an operation planner OPP, it receives the quality descriptors of the spraying (blue box) as inputs and delivers the necessary operating conditions including the diameter $d_{0}$ for the spraying system. The pressure and flow reference values for the control loop are delivered by the soft sensor as shown in the block diagram of Figure 1. In this case, the samples of the quality vector $\{Q_{i}\},i=\cdots ,N$ were used to mount the X data in Algorithm 1. The output of the soft sensor is the vector of operation conditions of the sprayer. In the model for the soft-sensor block, shown in Figure 1, the regression model based on PCA or *k* -NN can be selected.

### 2.5. Agricultural Sprayer Development System

To validate the developed soft sensor, the platform developed at the Brazilian Agricultural Research Corporation (Embrapa Instrumentation, São Carlos, SP, Brazil) in partnership with the School of Engineering of São Carlos University of São Paulo (EESC-USP), both in Brazil, was used. This platform was used for sprayer development analysis and operates as an agricultural sprayer development system (ASDS) using a National Instruments^®^ embedded controller, NI-cRIO^®^, which works on the LabView^®^ platform. The cRIO^®^ architecture integrates four components: a real-time processor, a user-programmable FPGA, modular I/O, and a complete software tool-chain for programming applications [28,29,30].

The ASDS is based of the boom sprayer hydraulic configuration and has an advanced development system that enables the design of architectures involving the connections of hydraulic components and devices, mechanical pumps, and electronic and computer algorithms, as illustrated in Figure 3.

Moreover, the system has hydraulic devices, which can be used to make any configuration of commercial agricultural sprays and new prototypes of sprayers, a user interface for system monitoring and control, and an electromechanical structure that emulates the movement of the agricultural sprayer in the field, as shown in Figure 4.

### 2.6. Data Collection

This section starts by describing the geometry and characteristics of the full cone nozzle as well as the displacement of the water-sensitive papers used to collect the data, which is followed by an analysis of the interpolation models used to increase the number of the samples used in the regression modelling PCA based.

#### 2.6.1. Full Cone Nozzle

The full cone nozzle is one of the most used nozzles in agricultural sprayers due to its constructive aspects, as it has a good uniformity of droplet spectrum and its geometry facilitates the development of analytical models [31]. The full cone nozzle is composed of three main parts: the entrance where pressurized liquid enters, a chamber responsible for generating turbulence in the liquid, and an outlet for which the function is to increase the liquid velocity and then to generate breaking drops in a circular footprint filled with liquid (Figure 5).

The nozzles used for tests and data collection were the full cone MAG CH model from the Brazilian company, Magnojet^®^. This nozzle is made using a technical ceramic core (99% alumina) to offer a high resistance to corrosive chemicals and a good application rate accuracy. This model of nozzle has a cone opening angle of 80${}^{\circ }$ and offers good coverage and penetration into crops [32]. The MAG CH model was selected with the endorsement of a specialist in this area, since it offers different sizes ranging from nozzles with fine drops (F) to nozzles with ultra-coarse drops (UC). In addition, the operating conditions of the ASDS (pressure, flow, application rate, and speed of application) were adjusted to give flexibility for testing several nozzles of the CH model with different droplet sizes.

Water-sensitive papers of size 7.68 cm${}^{2}$ (2.50 cm × 3.07 cm) were used to collect the drop size distribution pattern. This paper collects watermarks produced by the drops, which can be analyzed using a pattern recognition program to obtain the average diameters. A detailed diagram of the experimental setup is shown in Figure 6. The water-sensitive papers were displayed on an aluminum bar, with an impermeable paint coating that is positioned transversely to the movement of the application and spaced to collect all information from the drop distribution of all nozzles. The spraying was performed at a height of 51 cm. The distance between each nozzle was set to 50 cm (Figure 6) [33].

The water-sensitive papers were placed at critical points on the aluminum bar, which are to be considered in the distribution of median diameters. The critical points are taken beyond the nozzle cones (P1 and P9 in Figure 6) to collect data regarding the droplets with potential drifting. Two papers were placed at the external nozzles to collect the application pattern without overlap (P2 and P8 in Figure 6). Two water-sensitive papers were placed in the center of the overlapping between the nozzle cones (P4 and P6 in Figure 6), and three papers were placed in the center of the cones, perpendicular to the nozzle (P3, P5, and P7 in Figure 6).

#### 2.6.2. Positions of the Nozzles

It is important that the soft sensor considers the difference between observations obtained from different positions of the nozzles in the spray boom, i.e., it is important to consider the position of the nozzle to design the soft sensor and to obtain the results from the descriptors of quality in each position. Therefore, the position of the nozzle on the spray boom was added as another condition for the operation of agricultural machinery. This fact brings to the soft sensor a new dimension that helps to improve the efficiency and quality of the process because the softs sensor gives information on the quality descriptors with the best operating conditions and the position in the spray boom to get the best results. The consideration of the position of the nozzle along the spray boom is of great importance when making decisions related to the agricultural spraying process. Therefore, three critical positions of a spray nozzles were considered, as described in Figure 7.

The positions of the critical points are labeled as $p_{o}=1$ for the nozzles without overlapping (P2 and P8 in Figure 7), $p_{o}=2$ for the center of the overlapping of the nozzle cones (P4 and P6 in Figure 7), and $p_{o}=3$ for the center of the cones (P3, P5, and P7 in Figure 7). The P1 position is used as a reference point to evaluate the occurred errors in all positions.

To obtain the mean and median diameters using water-sensitive papers, the tool DropScope^®^ made by Ablevision^®^ was used. The data exploration, analysis of results, and construction of the soft-sensor were performed using the MATLAB^®^ and Simulink^®^ software.

#### 2.6.3. Collected Data and Interpolated Models

The experimental setup for collecting data for each tested nozzle are shown in Table 1, with $A_{p}$ [^*ℓ*^/ha] being the application dose rate and ΔP, $Q_{p}$ [^M^3^^/*s*], $V_{p}$ [^km^/*h*], and $d_{0}$ [mm] as already defined. Four conditions were tested, one per nozzle, with different discharge orifice diameters using models CH0.5, CH1, CH3, and CH6 from Magnojtet^®^. These nozzles were selected based on the recommendations from a specialist in the area of agricultural application to ensure a wide range of drop sizes within the database.

For each condition, there were 5 replicates, where the first 3 had the same operating conditions (S in Table 2). The fourth repetition was performed after a 10% lowering of sprayer boom pressure, and the fifth repetition was performed after a 10% increase in sprayer boom pressure. There were 9 water-sensitive papers positioned on the aluminum, and for each paper, two samples were collected. Thus, the total number of samples per repetition was 18, and the total samples for each condition, consisting of 5 repetitions, was 90. Through the 4 conditions, 360 samples were collected. The information collected experimentally from each water-sensitive paper was used to obtain the quality vector defined before as Q.

Data exploration was then performed on the collected data to analyse the data characteristics. Through a quartile-quartile plot (QQ plot), the close relationship between the quality descriptors and a normal curve was observed. Examples of the QQ plot applied to the data is shown in Figure 8. The results for the descriptors RA (Figure 8a) and VMD (Figure 8b) data are well accommodated in relation to the bisector line, which represents a high degree of agreement between them.

Based on the QQ plot results, it can be concluded that the quality descriptors are adherent to a Gaussian distribution. Therefore, to increase the amount of data for analysis, a Gaussian model was used for interpolation. Therefore, based on the experimentally collected training data, the final interpolation models were found, as shown in Figure 9, which is related to the evaluation of median diameters of the droplets. Furthermore, the models for the descriptors AR, CA, and RA can be observed in Figure 10a–c, respectively.

Moreover, in Figure 11, the obtained interpolation models for the operational conditions are shown. Based on the training values for the variables pressure ΔP [bar] and speed $V_{p}$ [km/h], it is possible to evaluate the usefulness of the soft-sensor models. In fact, the interpolation models, showed in Figure 11a,b, respectively, have been proven to be adequate for applications in agricultural machinery. Moreover, the orifice diameter of the discharge nozzles $d_{0}$ and their position on the spray boom $p_{o}$ have been used as the actually assembled, i.e., these values are related to their physical manufacturing characteristics.

Figure 11 shows the selected interpolation models for the quality descriptors related to the operating conditions. Based on the values obtained through training, it is possible to obtain both the optimized pressure ΔP (bar) and the application speed of the boom that is carried on an agricultural machinery in $V_{p}$ (km/h). Application of Gaussian interpolation model can be observed in Figure 11a,b. It is important to observe that the discrete nature of the nozzle’s discharge orifice diameter $d_{0}$, and their position on the spray boom $p_{o}$ are considered available from their manufacturers. Consequently, for these operating conditions, the interpolation process was not carried out.

It is important to clarify that the operating conditions $d_{0}$ and $p_{o}$ were used in conjunction with the other interpolated operating conditions ΔP and $V_{p}$ to build the soft sensor. The results were computed using 1000 samples for each descriptor.

### 2.7. Validation Methods

To present and analyze the results of the developed soft sensor, the following methods were used: control chart and error bars (MATLAB^®^), the root mean square error (RMSE), and and the correlation coefficient. The control chart was used to graphically compare the estimated and real values. The control chart calculates the upper and lower control limits (LCL/UCL) based on the process data and detects where undesirable changes occur in the process, based on the variation of data. The LCL and UCL are marked with red lines on the chart. Finally, error bars were used to observe the estimated values with greater or lesser error. The RMSE and the correlation coefficient denoted $C_{c}$ were calculated for each soft-sensor response.To validate the regression model, four repetitions of the experiments were used to obtain new data for the soft-sensor model, one for each nozzle model, shown in Table 1. Each repetition had 14 samples of water-sensitive paper, creating 56 new observations.

## 3. Results and Discussions

First, an exploration of the data was made. To apply techniques that work with maximization of variance, such as PCA or reduction of errors, it is important that the data of the random observations fit a normal curve. Therefore, a QQ plot was used to determine the fit of data to a normal distribution. Then, a Grubbs test was performed on the collected observations to detect possible outliers. The QQ plots and the Grubbs test are explained in [33].

### 3.1. PCA Soft Sensor Used as Quality Predictor

For the construction of the soft sensor as a predictor of the quality descriptors, the operating conditions given by vector *O*, defined in Section 2.3, were used to mount the data matrix X. Next, matrices A and Z, defined in Section 2.2.2, were calculated following Algorithm 1. For this soft-sensor model, three PCs comprise 100 % of the data variation and thus the dimensionality of the observations was taken as M = 3. Observations of each quality descriptor were used as a column vector *y* to compute the regression coefficients $\hat{\gamma }$. The regression coefficients estimated with the scores of the PCs are shown in Table 3 for the operating condition vector *O* and the quality descriptor vector *Q*, as already defined in Section 2.3.

The coefficients $\hat{\gamma }$ relate the quality descriptors to the operating conditions. Each column in Table 3 describes a regression model based on PCA for each quality descriptor.

### 3.2. PPQD Soft-Sensor Results

The statistical parameters, resulting from the prediction of the quality descriptors using a PPQD soft sensor with and without interpolation data are presented in Table 4. The quality descriptor vector denoted *Q* was defined in Section 2.3. The superiority of the statistical results with interpolation is noticeable.

The first quality descriptor to be analyzed is the SMD. The results obtained with the soft-sensor based on PCA for the SMD quality descriptor are shown in Figure 12. When observing the control chart (Figure 12), it can be observed that, for values used as a test in this soft-sensor for the range 60 μm <SMD<140 μm, the soft sensor for the SMD descriptor based on PCA presented a good estimate; however, for values outside this range, the LCL/UCL limits are exceeded. The deviation found in the estimate, given by RSME=22.39 μm, is considered small for agricultural spraying processes, where the randomness of variables is high and can vary with small changes in operating conditions.

The second quality descriptor to be analyzed is the VMD. The results obtained with the soft sensor based on PCA for the VMD quality descriptor are shown in Figure 13. Observing the control chart for the PCA-based soft sensor (Figure 13), high estimation errors occur for VMD>200 μm. As shown in the control chart (Figure 13), for values greater than 200 μm, the process is out of control and the magnitude of the error bars is high. Therefore, the PCA-based soft sensor for the VMD presents a suitable estimate for values in the range 180 μm <VMD<200 μm. When comparing the curve estimated by the PCA soft sensor (blue line in Figure 13) with real value curve (orange line in Figure 13), the estimator manages to correctly track the curve of actual values. The error bars for this soft sensor are small in magnitude, which is a good indicator of the efficiency of the estimator constructed for this descriptor. This fact is observed from the low deviation of estimated data versus actual observations, where RMSE=7.01 μm.

Following analysis of the quality descriptors, the third descriptor to be analyzed is $D_{0.9}$. The results obtained with the soft sensor based on PCA for the $D_{0.9}$ quality descriptor are shown in Figure 14. Inspecting the control chart for the PCA-based soft sensor (Figure 14), a good estimate of $D_{0.9}$ is observed for values less than 500 μm. Therefore, for tested values, the upper limit is 450 μm and higher values are considered, out of control. However, for values of less than 450 μm, the soft sensor has the best estimation efficiency, since error bars are small. Making the comparison of the curves of the estimated value with the real value, it is observed that, for values in the range 250 μm $<D_{0.9}<500$ μm, the soft sensor manages to correctly track the curve of real values, which indicates a satisfactory efficiency in the estimation of the $D_{0.9}$ descriptor. This fact is reaffirmed, having a deviation RMSE=45.26 μm.

The next descriptor to be analyzed is the $D_{0.1}$. The results obtained with the soft-sensor based on PCA for $D_{0.1}$ quality descriptor are shown in Figure 15. The control chart for the diameter $D_{0.1}$ shows that the PCA-based soft sensor offers the best estimates for the test values in the range 110 μm to 130 μm. This range is consistent with the drop diameter values found for the descriptor $D_{0.1}$ in practice. In addition, it is observed that few values estimated are considered out of control, which is a good indication that the estimator has a suitable estimate of the descriptor. If the curve estimated by the PCA soft sensor (blue line in Figure 15) is compared with the real values (orange line in Figure 15), it can be seen that the estimator manages to correctly track the curve of real values with small estimation error bars. The efficiency of estimation of the soft-sensor based on PCA is good, with RMSE=7.41 μm, which is low.

Next, we analyze the AR descriptor. The results obtained with the soft sensor based on PCA for the AR quality descriptor are shown in Figure 16. Observing the control chart (Figure 16) for the PCA approach, for the test sample values used, the soft sensor estimates the application rate well for values less than 50 L/min. The total value of the deviation for this PCA-based soft-sensor is RMSE = 10.41 L/min. This deviation value is suitable owing to AR being a descriptor that is most affected by the position of the nozzle, e.g., the AR is greater in the overlap of cones than in the center of a cone.

The results obtained from the soft-sensor based on PCA for the CA quality descriptor are shown in Figure 17. Inspecting the control chart (Figure 17) for the PCA-based soft-sensor for this descriptor, it can be said that the estimate of this soft sensor is suitable. For the test sample values used, the control chart of this soft sensor shows high values in the error bars for estimates of CA above 14 mm${}^{2}$. Therefore, these values are considered out of control, which indicates a low efficiency beyond this limit. The same happens for CA below 10 mm${}^{2}$, which is also out of control. For the set of example values used to test the soft sensor, for CA values in the range of 10 mm${}^{2}$< CA <15 mm${}^{2}$, the soft sensor presents the best estimation levels. This is verified through the observation that the curve of estimated values correctly tracks the curve of real values with small values in the estimation error bars and when observing the low value of the total deviation found as RMSE = 0.91 mm${}^{2}$.

Next is the RA descriptor. The results for this descriptor are shown in Figure 18. The control chart shows that the estimation curve (blue line in Figure 18) and the curve of real values (orange line in Figure 18) are near most values, so the soft sensor manages to correctly track the curve of real values. The effectiveness of the RA soft sensor becomes more evident when observing the deviation error, RMSE=0.02.

Next, the results found with the soft sensor built as an operation planner in the process (OPP) based on the PCA approach are presented.

#### PCA Soft-Sensor Used as OPP

To develop the soft sensor as an operation planner in the process, the observations of the quality descriptors were used to obtain the data matrix X. Again, matrices A and Z, as already defined, were calculated via Algorithm 1. In this OPP application, six PCs comprise 100% of the data variation and thus dimensionality of the observations is taken as M = 6. Observations of each operating condition were used as a column vector *y* to compute the regression coefficients $\hat{\gamma }$. The regression coefficients for the OPP soft sensor estimated with the scores of the PCs are shown in Table 5 for the quality descriptor vector *Q* and operating condition vector *O*, as already defined in Section 2.3.

The coefficients $\hat{\gamma }$ relate the operating conditions with the quality descriptors. Each column in Table 5 describes a regression model based on PCA for each required operating condition.

### 3.3. OPP Soft-Sensor Results

The OPP soft sensor as an operational process planner makes the prediction of the machinery operating conditions based on the input of quality descriptors of the application process. The statistical parameters, resulting from the prediction using the PCA-based soft sensor with and without interpolation models for each predicted operating condition can be observed in Table 6. As in the PPQD case, the superiority of the statistical results with interpolation is noticeable. The operating condition vector denoted *O* was defined in Section 2.3.

The results of the soft sensor constructed for the operating condition ΔP are shown in Figure 19. For the examples used as a test, the soft sensor created on the basis of PCA has large error bars in magnitude when the soft sensor tries to estimate the operating condition less than ΔP=2.4 bar. This fact is verified by the poor ability of the estimation curve (blue line in Figure 19) to track the real value curve (orange line in Figure 19). However, when the value of test pressure is close to ΔP=3.4 bar, there are smaller error bars than for the first condition tested. The total deviation of the error is represented by the value RMSE=0.72 bar, which is considered medium in magnitude, and it can be observed that, for the values tested, this soft-sensor has an acceptable estimation capacity.

The results of the soft sensor constructed for the operating condition $V_{p}$ (displacement velocity of the sprayer) are shown in Figure 20. When analyzing the control chart (Figure 20) of the PCA-based soft sensor for the descriptor $V_{p}$, suitable estimates are observed in the whole range. The control chart (Figure 20) presents small error bars for estimated values in the conditions range between $V_{p}=5$ km/h and $V_{p}=15$ km/h. The soft-sensor estimation curve (blue line in Figure 20) adequately follows the curve of real values (orange line in Figure 20), and this indicates the suitable level of estimation that the PCA-based soft sensor has for the operating condition $V_{p}$. This is confirmed by the small total deviation value of the errors RMSE=4.14 km/h.

The results for the soft-sensor constructed for the operating condition $d_{0}$ are shown in Figure 21. In the control chart (Figure 21), for the values used as test examples, the soft sensor has a low estimation level for values close to $d_{0}=1.0$ mm. The small values of error bars in Figure 21 for the operating conditions close to $d_{0}=0.3$ mm, $d_{0}=0.5$ mm, and $d_{0}=0.7$ mm indicate that, for these values, the soft-sensor has a good estimation, and for values between those limits, i.e., for 0.3 mm <$d_{0}$< 1.0 mm. These facts are evident when one observes that the estimation curve (blue line in Figure 21) follows the curve of real values (orange line in Figure 21). In addition, the value of the total deviation of the error, represented by RMSE=0.35 mm, is classified as medium and indicates an acceptable level of estimation. Figure 21. Soft-sensor response (control chart and error bars) for the operating condition $d_{0}$. The low estimation level for values close to $d_{0}=1.0$ mm of such an operating condition is noticeable.

Finally, the operating condition to be analyzed in such an arrangement is position $p_{0}$ of the nozzle along the sprayer boom, for which the results are shown in Figure 22. Here, the control chart has been estimated by using PCA and the operating condition $p_{0}$. The error bars have values smaller than 9.32%, which confirms the acceptable estimation level for the operating condition $p_{0}$. The total deviation of the error for this soft sensor is RMSE=2.33 cm, which is low, considering that the distance between each position considered was about 25 cm. Therefore, such results indicate that the soft sensor based on the use of PCA actuated as a good estimator for the nozzle’s position in the sprayer’s boom.

### 3.4. *k*-NN Soft Sensor

The construction of the soft sensor as a predictor of process quality descriptors and operation planing based on the *k*-NN regression method requires forming a prediction data matrix X. In this matrix of numerical data, each row is an observation of the process and each column is a characteristic or variable of prediction *x*. In the case of the PPQD, the operating conditions in the agricultural spraying process (vector *O*) were used as predictor variables. On the contrary, in the case of the OPP soft sensor, the quality descriptors (vector *Q*) were used as predictor variables. On the other hand, as an target function or class label, for the PPQD soft sensor, each of the quality descriptors was used and, for the OPP soft sensor, each of the operation conditions was used as a target function. Thus, for each quality descriptor and each operating condition, a *k*-NN model was constructed.

To find the best distance function and the best value of nearest neighbors *k*, the *k*-NN classifier optimizer (*fitcknn*) was used. The results of this automatic optimization of hyper-parameters, for the PPQD and OPP soft-sensor, are presented in Table 7 and Table 8, respectively.

### 3.5. Comparative Results

For comparison purposes, in Table 9 and Table 10, the statistics of the PC and k− NN regression are given. It can be observed that the PPQD soft-sensor based on the PC regression offers better results. It is observed that, for quality descriptors, the value of the RMSE is quite high for the regression *k*-NN when compared to PC regression. This statistic parameter is an indication that, in this case, the PCA-based approach adequately estimates the quality descriptors, which is corroborated by the correlation coefficient $C_{c}$ value. It is important to note that, for the average diameters SMD, VMD, $D_{0.1}$, and $D_{0.9}$, which define the spectrum of drops, the difference in the errors between PC and *k*-NN regressions can be above 150 μm. For example, in the case of the mean diameter DMV for the approach *k*-NN, the value of the RMSE = 499.93 μm is extremely high. This amount of deviation in an actual application can generate phytosanitary problems in the cultures. Also, it is observed that, for quality descriptors CA and RA, which offer information on the uniformity of the application, the estimation efficiency of the soft sensor based on the PCA regression is higher when compared to the *k*-NN regression. Therefore, the PCA soft-sensor is more efficient and works better for making decisions about uniformity in a real application. Regarding the volume applied to the culture, represented by the AR application rate, the best estimate is also made by the PCA soft sensor; it is observed that the estimate made with *k*-NN may have deviations around 70 L/ha. This value is very large, which can lead to overapplication or underapplication problems in a real application.

On the other hand, regarding the OPP operation planner, the results between the regressions based on PCA or *k*-NN approaches, are observed to be close in efficiency except the $V_{p}$, which is better estimated by the PCA approach. In real applications, the application velocity is directly related to the volume applied to the cultures. The application velocity $V_{p}$ is a variable that has high impact on the quality of the real application to cultures. Therefore, having a soft sensor that offers adequate information to make a decision in the planning of operations is essential to reduce application errors. In this case, the soft-sensor based on PCA regression is more efficient to obtain adequate estimates of the operating conditions and to, thus, apply corrections in the planning of operations.

### 3.6. Implementation of the PCA Soft Sensor

As a result of the analysis presented, the implementation of the soft-sensor based on PC regression was performed as in Figure 23. The construction of the soft sensor thus begins with the entry of the historical data corresponding to the training matrices of quality descriptors $X_{Q_{i}}$ and operating conditions $X_{O_{i}}$. Next, data exploration is carried out in order to recognize the nature and to detect the possible outliers of the data. Identifying the nature of the data history, an interpolation is performed to increase the amount of data to analyze and to develop the soft-sensor. Then, the execution comes down to the choice of the type of output information required from the soft sensor, and the coovariance matrix as well as its representative eigenvectors and eigenvalues, are computed. Finally, the soft sensor delivers the information required for each case by executing the procedures presented in the flow chart in Figure 23 and explained in more detail in Section 2.3.

## 4. Conclusions

The obtained results from the proposed soft-sensor based on a PC regression model for the estimation of the spray quality descriptors PPQD showed reliable results. In addition, the soft-sensor results obtained with a practical application showed its strength in estimating the vector of quality descriptors. Moreover, the combination of constructed models enabled us to establish a relationship between the quality descriptors and the operating conditions for agricultural sprayers, considering the real applications in pest management.

The developed models consider the variability that droplet size may have in response to minor changes in the operating conditions, providing a useful tool for real-time decision making and more precise control application, as the proposed soft sensor can provide the best operating conditions for working with a nozzle on a desired position on the application bar. There is currently no instrument that can measure the quality of the application being made and that can determine the operating conditions that each nozzle should have at each particular spray bar position in real time. This fact largely justifies the use of the soft sensor built for real applications. In addition, there are currently individually controlled spray nozzles, and a tool that automatically determines application quality for each nozzle position on the spray bar can considerably eliminate application errors.

Therefore, based on this innovative strategy, it is possible to perform periodic evaluations of the quality of the application rate and to provide corrective actions to the operating conditions of sprayers to regulate variables, such as pressure, flow, and to select the appropriate nozzle and the desired specifications for the sprayer bar.

The obtained results with the soft-sensor based on a PC regression model for the operation planning OPP, demonstrated its capability to estimate appropriate operating conditions for agricultural rate of application. These results help to improve the quality of application, as it was possible to obtain the necessary information to configure the sprayer operating conditions a priori and to obtain higher levels of spraying quality, which is desirable for both agriculture and environmental protection.

From the results obtained with both regression methodologies, it can be concluded that the soft sensor based on the PC regression offers better estimation results for the quality descriptors as well as for the operating conditions of agricultural machinery. Therefore, with the soft-sensor based on PC regression, there is adequate information in decision-making processes in real time for the application of pesticides in spray form. Thus, corrective measures can be applied to improve the quality of the applications, considerably reducing the biological impact and the economic cost in this type of agricultural process.

In a future work, the implementation of embedded soft-sensors in customized agricultural devices will be considered. This will aggregate intelligence in the agricultural machinery sector, allowing for the connectivity of a soft-sensor device to controller area network environments. It is expected that the soft-sensor models built in this work can be considered not only for drift studies but also for the evaluation of other nozzle’s types. A possible way to carry out such future studies could be the use of advanced estimation algorithms based on neural networks (NN) or even on the support vector machine (SVM), which would provide the needed non-linearity and flexibility to better fit the estimators to measured data.

## Figures and Tables

**Figure 1 sensors-21-01269-f001:**
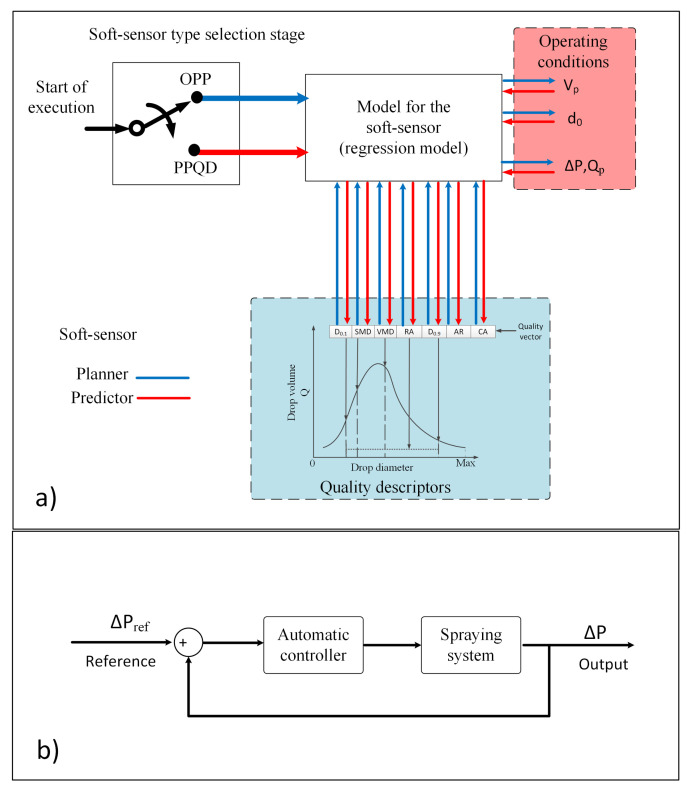
(**a**) Architecture of the soft sensor used as a predictor for the variables of the quality descriptors (red arrows) or operation planner of the agricultural spraying process (blue arrows). (**b**) Automatic system for the sprayer boom pressure.

**Figure 2 sensors-21-01269-f002:**
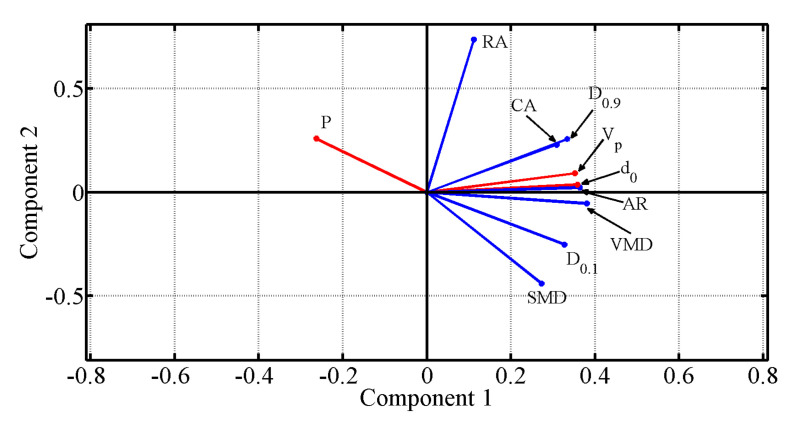
Biplot of the principal components (PCs) for application quality descriptors and operating conditions.

**Figure 3 sensors-21-01269-f003:**
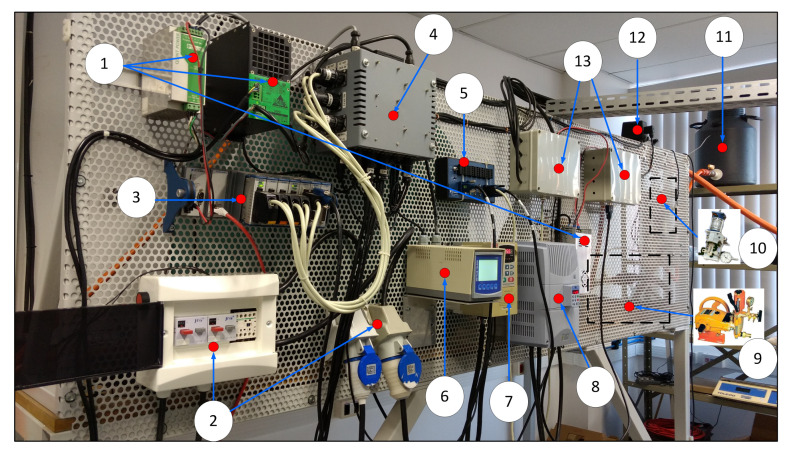
Front view of the agricultural sprayer development system (ASDS) electro-hydraulic devices: (1) power supplies, (2) electrical protection circuits, (3) modules for automation and control of input and output variables, (4) box with electronic circuits for signal conditioning, (5) CAN network bus, (6) transmitter for analog sensors, (7) frequency inverter for control of the spray pump, (8) frequency inverter for the control of industrial belt that simulates tractor movement in relation to sprayers, (9) spray pump, (10) two piston pumps for the injection of pesticides, (11) pesticide reservoir tank, (12) proportional valves for pressure and flow control, and (13) valve actuation circuits operated via CAN network.

**Figure 4 sensors-21-01269-f004:**
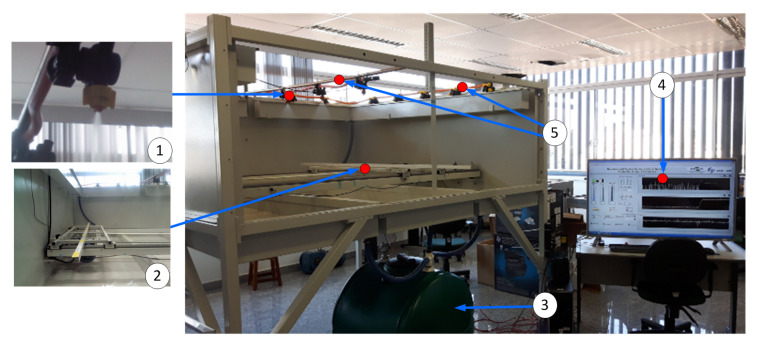
Development system for projects dedicated to the application of liquid agricultural inputs: (1) spray nozzle, (2) system that emulates the movement of the sprayer, (3) pesticide disposal tank, (4) user interface for the development system, and (5) spray booms.

**Figure 5 sensors-21-01269-f005:**
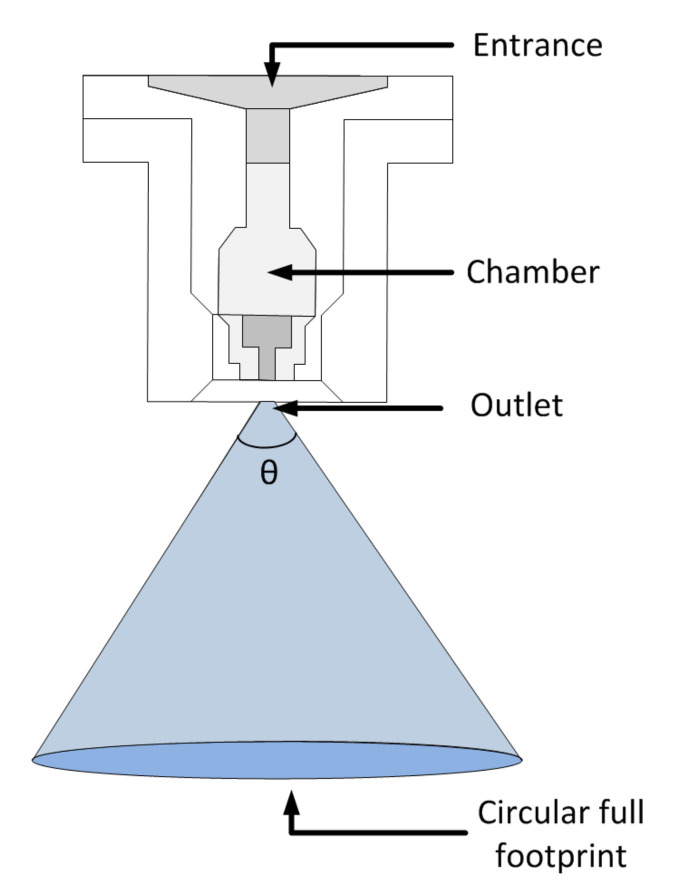
Full cone nozzle (adapted from the Magnojet^®^ catalog).

**Figure 6 sensors-21-01269-f006:**
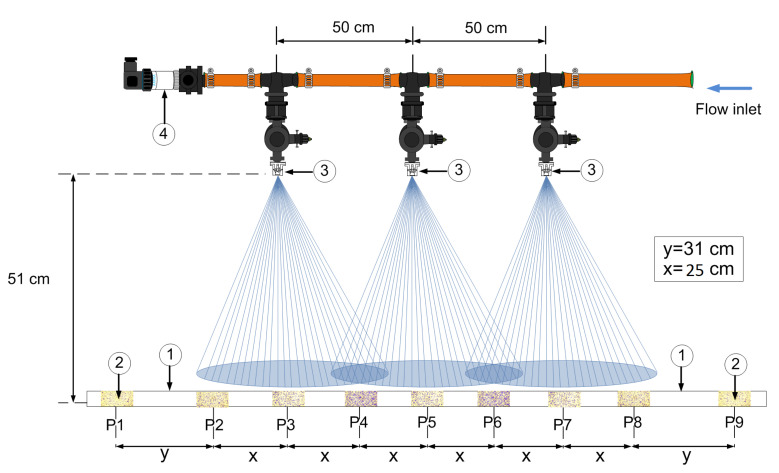
Magnified view of the spray boom with the nozzles used for data collection with (1) an aluminum bar with an impermeable paint coating, (2) water-sensitive papers, (3) a set of nozzles, and (4) a pressure sensor.

**Figure 7 sensors-21-01269-f007:**
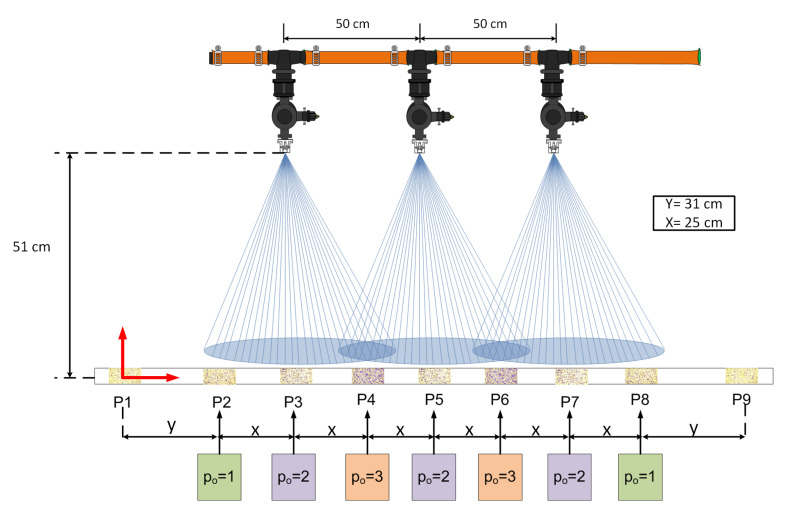
Labels of the critical points in the spray boom.

**Figure 8 sensors-21-01269-f008:**
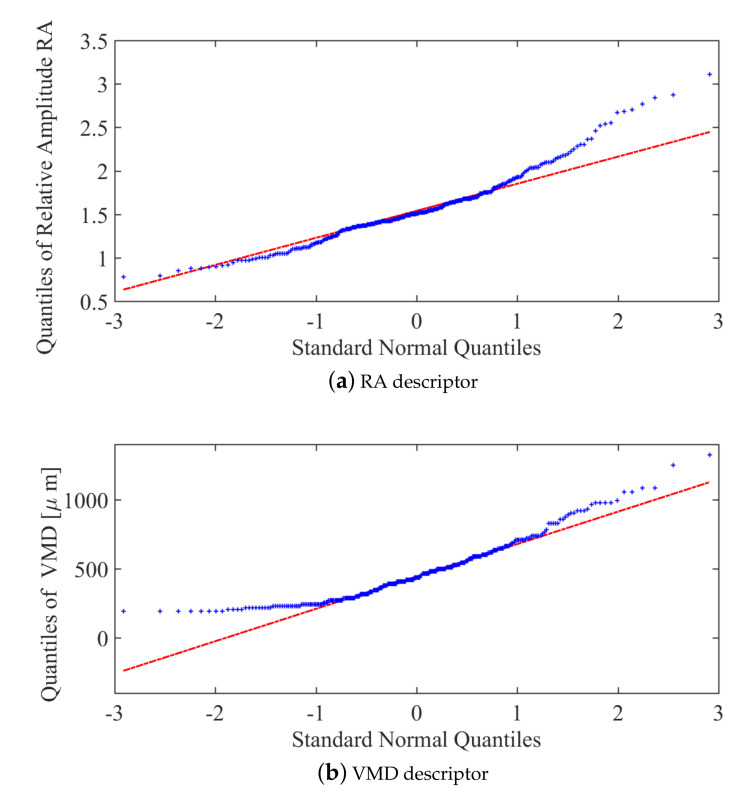
Quartile-quartile plot versus standard normal quantiles.

**Figure 9 sensors-21-01269-f009:**
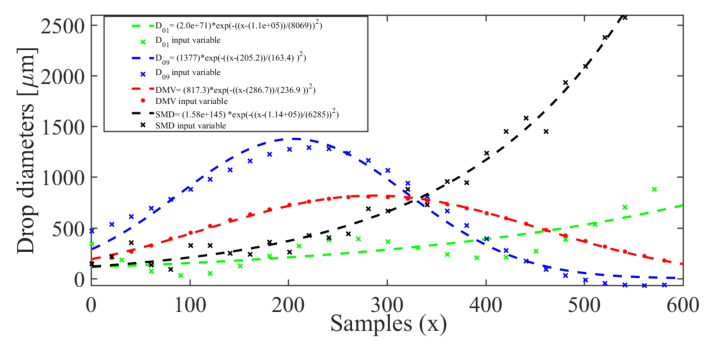
Gaussian curves of interpolation for mean and median diameter descriptors.

**Figure 10 sensors-21-01269-f010:**
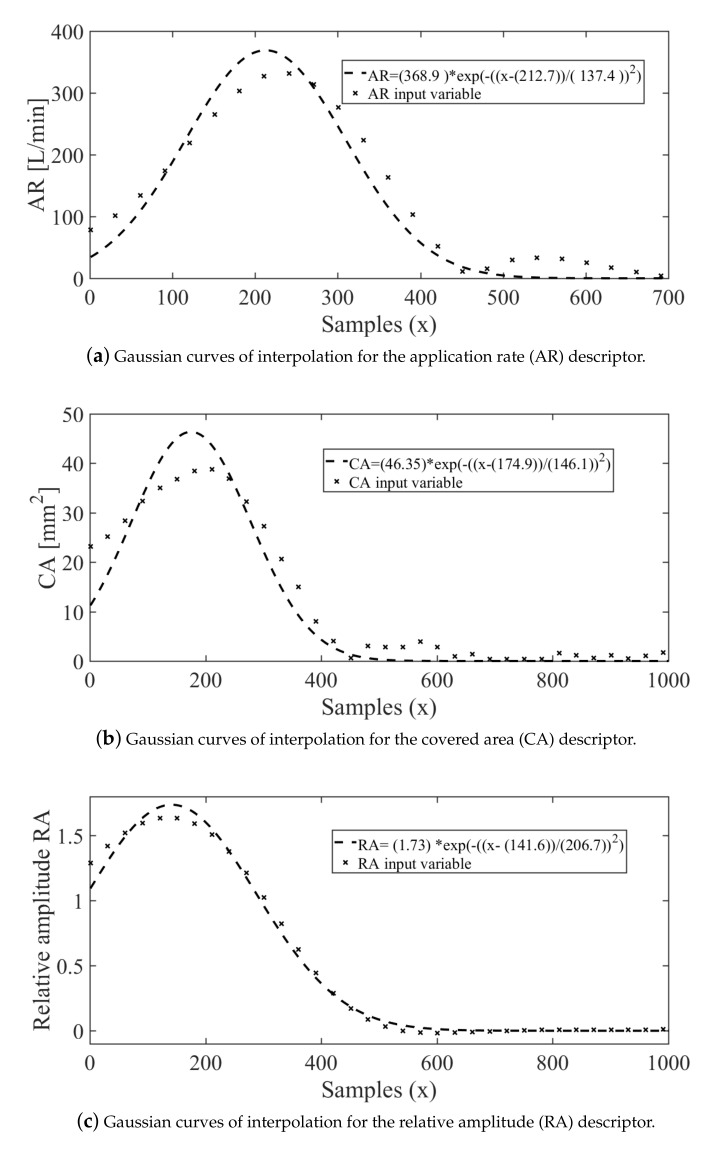
Models for the interpolation of the quality descriptors based on the Gaussian distribution.

**Figure 11 sensors-21-01269-f011:**
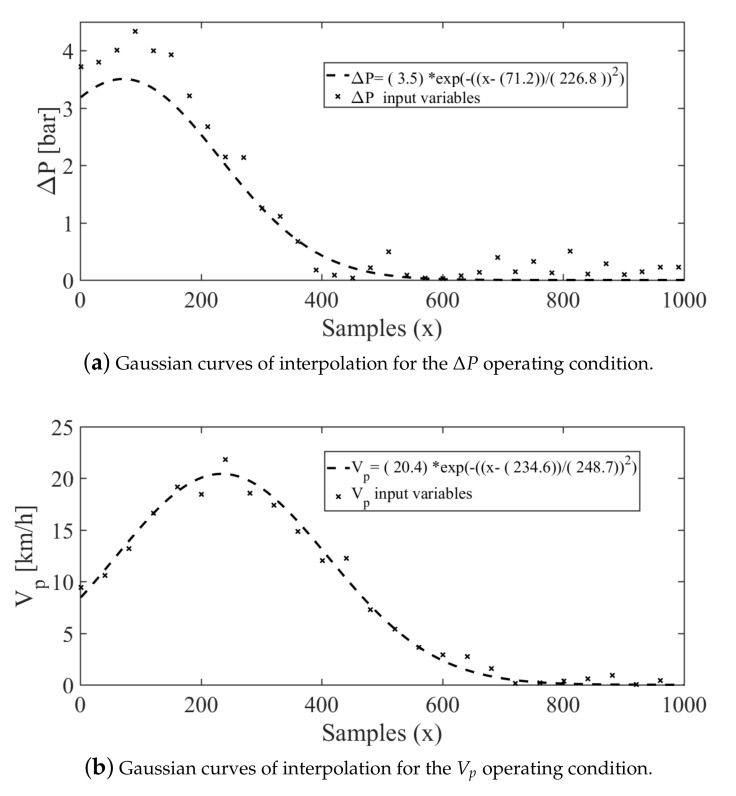
Interpolation results obtained for the operating conditions.

**Figure 12 sensors-21-01269-f012:**
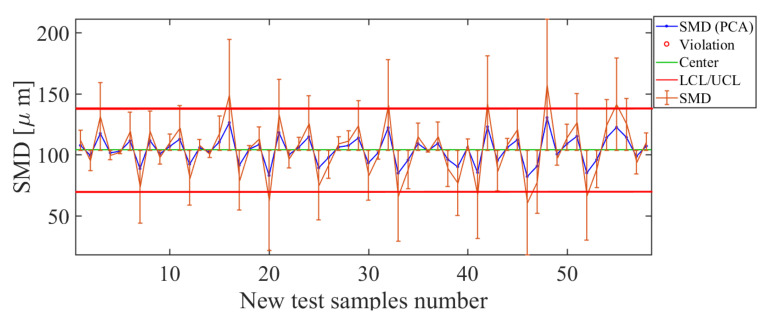
Soft-sensor response (control chart and error bars) for the Sauter mean diameter (SMD) descriptor. For values of SMD>140 μm and SMD<60 μm, high estimation error levels are observed.

**Figure 13 sensors-21-01269-f013:**
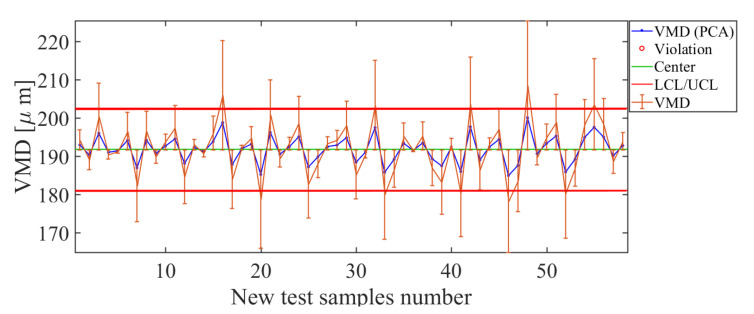
Soft-sensor response (control chart and error bars) for the volumetric median diameter (VMD) descriptor. For values in the range 180 μm <VMD<200 μm, suitable estimates are provided.

**Figure 14 sensors-21-01269-f014:**
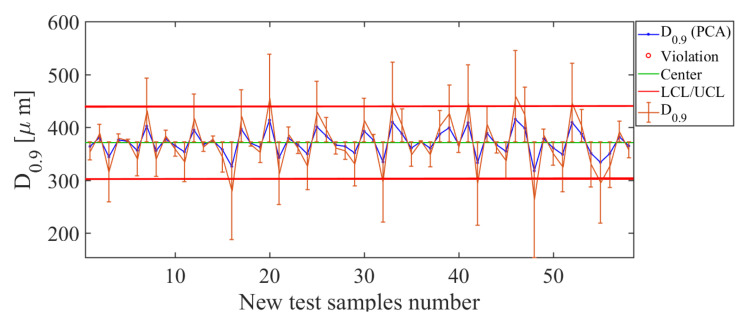
Soft-sensor responses (control chart and error bars) for the $D_{0.9}$ descriptor. For values in the range 300 μm $<D_{0.9}<500$ μm, it is observed that the soft sensor has the best estimation efficiency, since small error bars are observed.

**Figure 15 sensors-21-01269-f015:**
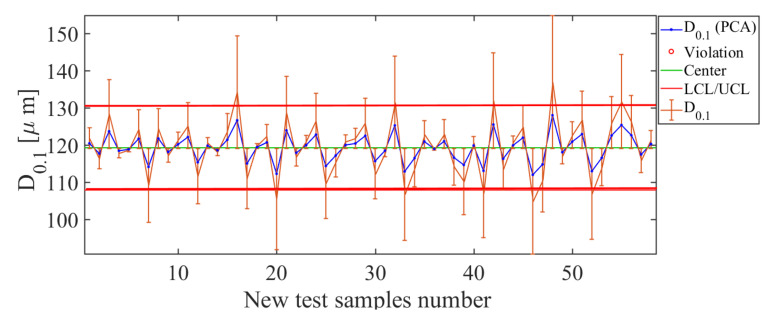
Soft-sensor responses (control chart and error bars) for the $D_{0.1}$ descriptor. Best estimates are in the range 100 μm to 135 μm, which is consistent with the drop diameter values that is found in practice.

**Figure 16 sensors-21-01269-f016:**
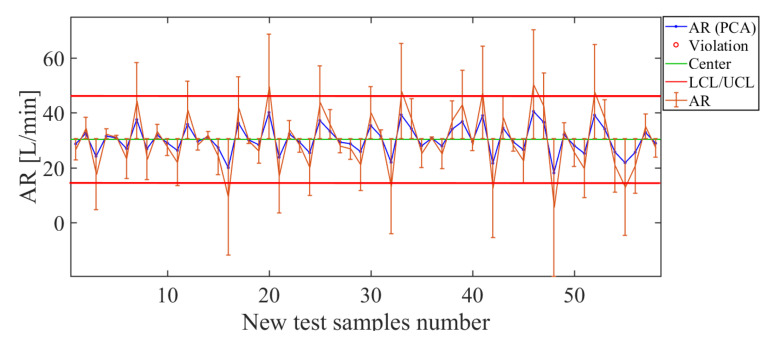
Soft-sensor response (control chart and error bars) for the AR descriptor: for values less than 50 L/min, suitable estimates of the application rate are given.

**Figure 17 sensors-21-01269-f017:**
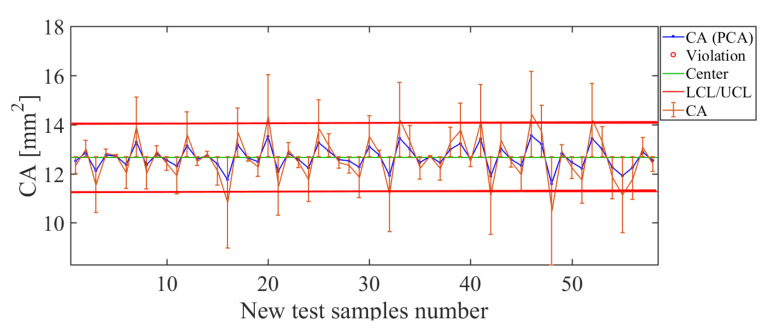
Soft-sensor response (control chart and error bars) for the CA descriptor. The best estimation levels are obtained in the range 10 mm${}^{2}$< CA < 15 mm${}^{2}$.

**Figure 18 sensors-21-01269-f018:**
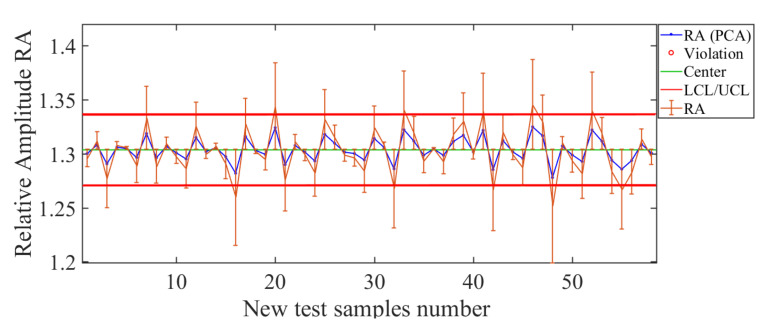
Soft-sensor response (control chart and error bars) for the RA descriptor. Suitable estimates are observed in the tested whole range 1.27<RA<1.35.

**Figure 19 sensors-21-01269-f019:**
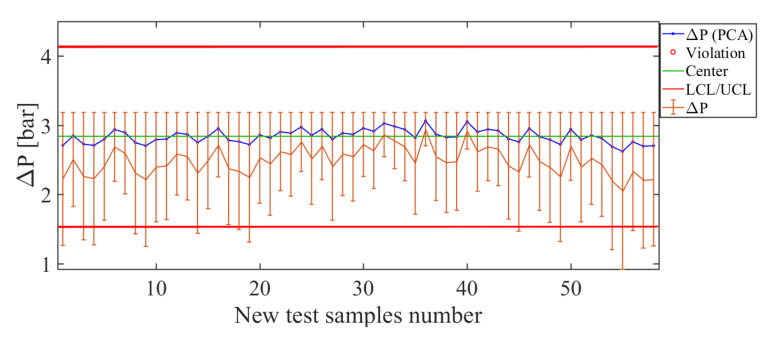
Soft sensor response (control chart and error bars) for the operating condition ΔP. The magnitudes of the error bars are smaller close to the condition ΔP=3.0 bar.

**Figure 20 sensors-21-01269-f020:**
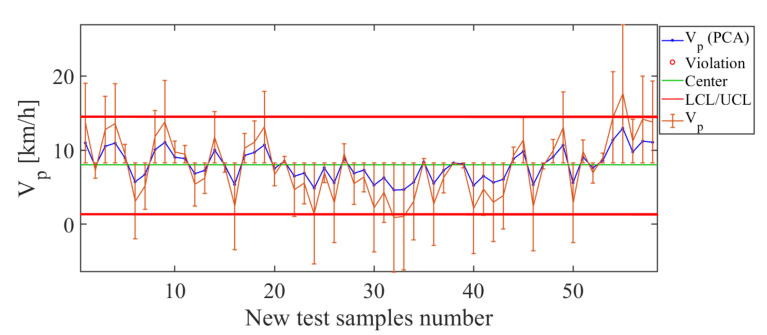
Soft-sensor response (control chart and error bars) for the operating condition $V_{p}$. Note the low level of the estimates for values less than $V_{p}=5$ km/h.

**Figure 21 sensors-21-01269-f021:**
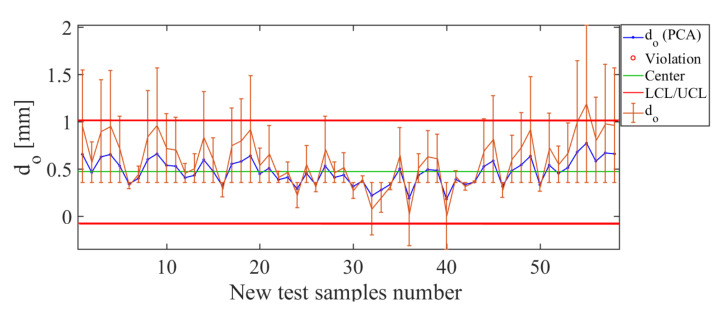
Soft-sensor response (control chart and error bars) for the operating condition $d_{0}$. Note the low estimation level for values close to $d_{0}=1.0$ mm of this operating condition.

**Figure 22 sensors-21-01269-f022:**
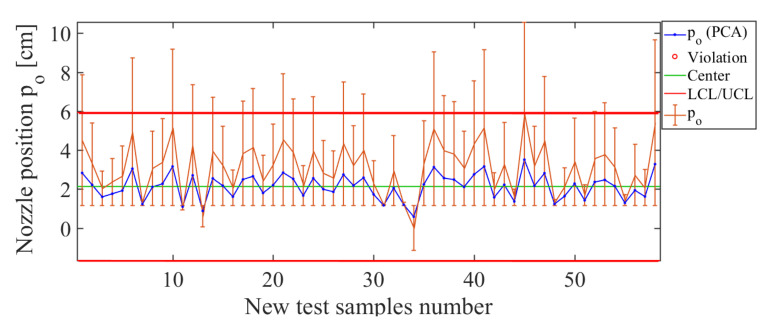
Soft-sensor response for the operating condition $p_{0}$. Soft-sensor response (control chart and error bars) for the operating condition $p_{0}$. Note the low estimation level for values greater than $p_{0}=6$ cm.

**Figure 23 sensors-21-01269-f023:**
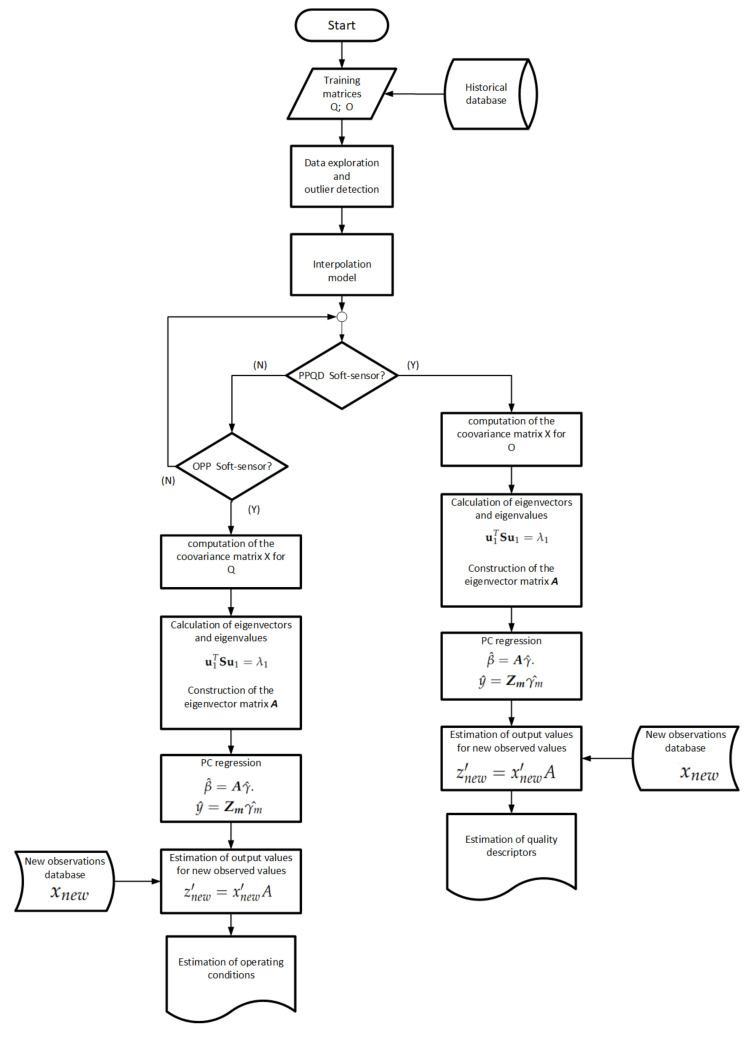
Flow chart stages of the proposed soft sensor to obtain estimates of the quality descriptors and the operating conditions.

**Table 1 sensors-21-01269-t001:** Experimental setup for the ASDS using full cone nozzles (Magnojet^®^).

	Nozzle	Drop Pattern	ΔP	$Q_{p}$	$A_{p}$	Temp	Humidity	$V_{p}$	$d_{0}$
			[bar]	[L/min]	[L/ha]	[°C]	[%]	[km/h]	[mm]
1st	CH0.5	Fine	3.4	0.53	67	23.6	51	10	0.5
2nd	CH1	Medium	3.4	1.02	85	23.4	61	14	1.0
3rd	CH3	Coarse	3.4	1.46	100	24.0	49	18	1.5
4th	CH6	Ultra Coarse	2.4	1.90	120	23.7	58	20	2.0

**Table 2 sensors-21-01269-t002:** Arrangement of samples for each condition.

	Nozzle	N° Repetition			Total	N° Papers	N° Samples	Total Samples
		S	−10%	+10%				
1st	CH0.5	3	1	1	5	9	18	90
2nd	CH1	3	1	1	5	9	18	90
3rd	CH3	3	1	1	5	9	18	90
4th	CH6	3	1	1	5	9	18	90
						Total collected samples		360

**Table 3 sensors-21-01269-t003:** Estimated regression predictor of process quality descriptors (PPQD) coefficients $\hat{\gamma }$.

$\hat{\gamma }$	CA	AR	RA	$D_{0.1}$	SMD	VMD	$D_{0.9}$
*P*	0.59	0.62	0.60	−0.49	−0.42	0.55	0.63
$V_{p}$	0.53	−0.12	0.54	−0.12	−0.01	−0.52	0.00
$d_{0}$	−1.28	−3.40	−1.27	3.70	6.10	9.56	−2.48
$p_{os}$	−0.75	−2.55	−0.75	−3.85	−4.04	1.65	−1.85

**Table 4 sensors-21-01269-t004:** The statistical results of the PPQD soft-sensor prediction with and without a interpolation model.

	μ		σ		RMSE		$C_{c}$	
	**with**	**without**	**with**	**without**	**with**	**without**	**with**	**without**
SMD [μm]	104.51	272.78	22.55	81.01	22.39	148.23	0.73	0.65
VMD [μm]	192.00	472.85	7.06	78.25	7.01	130.046	0.95	0.84
$D_{0.9}$ [μm]	369.83	490.32	45.60	233.38	45.26	346.27	0.83	0.76
$D_{0.1}$ [μm]	119.45	173.55	7.46	30.45	7.41	56.34	0.75	0.62
CA [cm${}^{2}$]	12.64	33.51	0.92	9.96	0.91	11.09	0.98	0.66
AR [L/ha]	26.40	108.33	4.31	40.12	6.00	66.97	0.76	0.70
RA	1.30	1.60	0.03	0.30	0.02	0.40	0.97	0.22

**Table 5 sensors-21-01269-t005:** Estimated regression operational process planner (OPP) coefficients $\hat{\gamma }$.

	ΔP	$V_{p}$	$d_{0}$	$p_{\mathrm{os}}$
CA	0.59	0.40	0.40	0.34
AR	0.21	0.05	0.05	−0.16
RA	−1.07	0.38	0.38	−0.15
$D_{0.1}$	−0.91	−0.21	−0.20	−0.40
SMD	−0.60	0.42	0.42	−0.45
VMD	−1.75	1.24	1.24	−1.04
$D_{0.9}$	−1.53	−1.09	−1.09	−3.80

**Table 6 sensors-21-01269-t006:** Statistical results of OPP soft-sensor prediction with and without a interpolation model.

	μ		σ		RMSE		$C_{c}$	
	**with**	**without**	**with**	**without**	**with**	**without**	**with**	**without**
ΔP [bar]	2.50	3.15	0.20	0.36	0.72	0.76	0.35	0.30
$V_{p}$ [km/h]	7.75	15.50	4.10	5.26	4.14	7.32	0.83	0.78
$d_{0}$ [mm]	0.59	1.25	0.27	0.31	0.35	0.40	0.54	0.42
$p_{0}$ [cm]	3.11	3.58	1.29	2.40	2.34	8.75	0.64	0.47

**Table 7 sensors-21-01269-t007:** *k* Neighbors of the PPQV soft sensor found using optimization.

	Num.	Distance	Eval. Time	Obj. Value
	Neighbors		[Seg]	
SMD	11	Mahalanobis	0.10	0.94
VMD	1	Hamming	0.06	0.82
$D_{0.9}$	2	Std. Euclidean	0.06	0.91
$D_{0.1}$	6	Cosine	0.05	0.63
AR	18	Euclidean	0.07	0.99
CA	6	Hamming	0.07	0.97
RA	18	Std. Euclidean	0.06	0.93

**Table 8 sensors-21-01269-t008:** *k* neighbors of the OPP soft-sensor found using optimization.

	Num.	Distance	Eval. Time	Obj. Value
	Neighbors		[Seg]	
ΔP [bar]	17	Mahalanobis	0.05	0.53
$V_{p}$ [km/h]	14	jaccard	0.06	0.98
$d_{0}$ [mm]	6	Chebychev	0.07	0.22
$p_{0}$ [cm]	8	Mahalanobis	0.06	0.29

**Table 9 sensors-21-01269-t009:** Statistics of the PPQD soft-sensor predictor with PC regression compared to the *k*-NN regression.

	μ		σ		RMSE		$C_{c}$	
	**PCA**	k **-NN**	**PCA**	k **-NN**	**PCA**	k **-NN**	**PCA**	k **-NN**
SMD [μm]	104.51	208.67	22.55	115.77	22.39	187.43	0.73	0.33
VMD [μm]	192.00	445.93	7.06	95.78	7.01	171.14	0.95	0.45
$D_{0.9}$ [μm]	369.83	943.54	45.60	306.36	45.26	499.93	0.83	0.40
$D_{0.1}$ [μm]	119.45	164.99	7.46	22.92	7.41	39.18	0.75	0.49
CA [cm${}^{2}$]	12.64	29.53	0.92	8.46	0.91	12.27	0.98	0.75
AR [L/ha]	26.40	73.80	4.31	42.63	6.00	74.22	0.76	0.42
RA	1.30	1.47	0.03	0.29	0.02	0.46	0.97	0.39

**Table 10 sensors-21-01269-t010:** Statistics of the OPP soft-sensor predictor with PC regression compared to *k*-NN regression.

	μ		σ		RMSE		$C_{c}$	
	**PCA**	k **-NN**	**PCA**	k **-NN**	**PCA**	k **-NN**	**PCA**	k **-NN**
ΔP [bar]	2.50	3.17	0.20	0.17	0.72	0.66	0.35	0.42
$V_{p}$ [km/h]	7.75	20.93	4.10	11.40	4.14	26.60	0.83	0.19
$d_{0}$ [mm]	0.59	1.18	0.27	0.13	0.35	0.27	0.54	0.61
$p_{0}$ [cm]	3.11	1.92	1.29	0.36	2.34	0.74	0.64	0.86

## Data Availability

The data presented in this study are available on request from the corresponding author.

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
