# Peer review of "Using Soft Sensors as a Basis of an Innovative Architecture for Operation Planning and Quality Evaluation in Agricultural Sprayers†"

_sensors, 2021, doi:10.3390/s21041269_

Round 1

Reviewer 1 Report

This paper deals with soft-sensor for operation planning and quality evaluation in agricultural sprayers. The topic is relevant and the idea is interesting. Here are some suggestions for the paper.

1 In the abstract and introduction, the authors should pay more attention to emphasis the motivations and contributions of this paper.

2. There are SOME notations without proper definition and explanation in section 3. The authors should carefully exam them.

3 For the case study, it is better to compare the proposed method with some widely used prediction techniques.

4.Apart from these traditional techniques, there are very recent works on deep learning based soft sensors. The authors should mention the most recent references.

5. The references should be formed with the standard of this journal.

6. What are the limitations and future directions for the present work. They can be briefly given them in the conclusion part.

Reviewer 2 Report

The manuscript focuses essentially on an application of well known data-driven approach. More specifically, linear models are obtained by using the PCA. The obtained models are eventually verified. 

Though the topic can be of interest for readers with specific interests in agriculture, there is no novelty for Sensor's readers. 

My suggestion is that the manuscript is submitted to a more focused journal or to a conference.

Reviewer 3 Report

The proposed application is interesting, some efforts could be done to improve the paper, as listed below:

-the methodology is not innovative, being the very basic development of data-drive SS, what is new is the application, so I suggest to modify the paper title;

-the abstract is not clear, in particular as regards the SS purposes and I/O variables;

-the state-of-the-art should be improved and could be better organized i.e. could discuss separately the applications, with emphasis to those relevant to the proposed one,  the methodologies, including i.e. deep neural networks, DBNs and other semisupervised approaches and the papers specific to the SS design procedure, i.e. to the input selection procedure;

-in line 82 the authors claims the use of computational intelligence, however their method is a classic statistical one;

-more attention should be paid to the relevance of the SS in the specific application and to the related state-of-the art;

-what about the SS design without data interpolation? a comparison would be interesting;

-plots of the input variables, with marks on the interpolated data, should be reported;

-please, evaluate also the correlation coefficient between actual and estimated outputs and the statistical properties of the residuals;

-the choice of the methods, with respect to many other SS design procedure, should be justified; is it possible to compare it with another method (e.g. a nonlinear one)?

Round 2

Reviewer 2 Report

The Authors have significantly improved the content of the manuscript that can be now accepted

Reviewer 3 Report

The authors revised the paper according to my comments.